# *Lactobacillus acidophilus* Expressing Murine Rotavirus VP8 and Mucosal Adjuvants Induce Virus-Specific Immune Responses

**DOI:** 10.3390/vaccines11121774

**Published:** 2023-11-28

**Authors:** Darby Gilfillan, Allison C. Vilander, Meichen Pan, Yong Jun Goh, Sarah O’Flaherty, Ningguo Feng, Bridget E. Fox, Callie Lang, Harry B. Greenberg, Zaid Abdo, Rodolphe Barrangou, Gregg A. Dean

**Affiliations:** 1Department of Microbiology, Immunology and Pathology, College of Veterinary Medicine and Biomedical Sciences, Colorado State University, Fort Collins, CO 80523, USA; darbygil@colostate.edu (D.G.); allison.vilander@colostate.edu (A.C.V.); bridget.fox@aims.edu (B.E.F.); callie.lang@colostate.edu (C.L.); zaid.abdo@colostate.edu (Z.A.); 2Department of Food, Bioprocessing and Nutrition Sciences, North Carolina State University, Raleigh, NC 27695, USA; mpan6@ncsu.edu (M.P.); jgoh@agbiome.com (Y.J.G.); sjoflahe@ncsu.edu (S.O.); rbarran@ncsu.edu (R.B.); 3Departments of Medicine and Microbiology and Immunology, School of Medicine, Stanford University, Stanford, CA 94305, USAhbgreen@stanford.edu (H.B.G.); 4VA Palo Alto Health Care System, Department of Veterans Affairs, Palo Alto, CA 94304, USA

**Keywords:** rotavirus, *Lactobacillus acidophilus*, next-generation vaccine

## Abstract

Rotavirus diarrhea-associated illness remains a major cause of global death in children under five, attributable in part to discrepancies in vaccine performance between high- and low-middle-income countries. Next-generation probiotic vaccines could help bridge this efficacy gap. We developed a novel recombinant *Lactobacillus acidophilus* (rLA) vaccine expressing rotavirus antigens of the VP8* domain from the rotavirus EDIM VP4 capsid protein along with the adjuvants FimH and FliC. The *upp*-based counterselective gene-replacement system was used to chromosomally integrate FimH, VP8Pep (10 amino acid epitope), and VP8-1 (206 amino acid protein) into the *L. acidophilus* genome, with FliC expressed from a plasmid. VP8 antigen and adjuvant expression were confirmed by flow cytometry and Western blot. Rotavirus naïve adult BALB/cJ mice were orally immunized followed by murine rotavirus strain EC_WT_ viral challenge. Antirotavirus serum IgG and antigen-specific antibody-secreting cell responses were detected in rLA-vaccinated mice. A day after the oral rotavirus challenge, fecal antigen shedding was significantly decreased in the rLA group. These results indicate that novel rLA constructs expressing VP8 can be successfully constructed and used to generate modest homotypic protection from rotavirus challenge in an adult murine model, indicating the potential for a probiotic next-generation vaccine construct against human rotavirus.

## 1. Introduction

Diarrheal illness is the fourth leading cause of death in children under five years of age worldwide [1]. It is estimated that rotavirus infection killed approximately 128,000 children in 2016 and accounted for 19.11% of all diarrhea-associated deaths in 2019 [2,3]. The World Health Organization recommends including a rotavirus vaccine in all global vaccination protocols and, currently, 124 countries have introduced rotavirus into their national immunization programs [3,4]. There are currently four vaccines (Rotarix^®^, RotaTeq^®^, Rotavac^®^, and Rotasiil^®^) that are preapproved internationally, all of which are formulated as orally delivered attenuated live-virus vaccines [5]. However, these current rotavirus vaccines have limited efficacy (50–60%) in lower-middle-income countries (LMIC) with a low sociodemographic index and high childhood mortality [2,6,7]. This discrepancy in vaccine efficacy leaves millions of children at risk for rotavirus infection and diarrheal illness [8], and the cause of the decreased response to the attenuated live-rotavirus vaccines is poorly understood. Factors such as mal/undernutrition, impact of the intestinal microbiome, coinfection with enteric pathogens, environmental enteric dysfunction, and maternal antibody interference are believed to play a role [7,9,10]. Attenuated vaccines also have inherent risks, including reversion to virulence, recombination with circulating viruses, and decreased safety in immune-suppressed populations. Additionally, they require cold-chain storage and specialized personnel/training for administration [11]. Intussusception has also been a historic risk of live attenuated rotavirus vaccines [12]. It is therefore recognized that a new generation of vaccines is needed to overcome these many obstacles and provide improved protection for more people.

B-cells and IgA have been identified as key to mucosal protection against rotavirus infection in humans and animal models [13,14]. Mucosal immunization by orally delivered vaccines is the most effective means of inducing mucosal IgA; however, there is a paucity of oral vaccine platforms that have been proven to be safe and immunogenic [15,16,17]. Probiotic-based vaccines have promise as a mucosal immunization platform [18]. Probiotics are generally regarded as safe (GRAS) as they are often commensal residents of the human microbiome, and offer logistical advantages to both orally and parentally delivered vaccines due to low production cost, storage sans cold chain, and ease of administration [12,19,20]. To date, *Lactobacillus* spp. have primarily been used as an adjuvant strategy for the current rotavirus vaccines, but the application of *Lactobacillus* as a vaccine platform itself is an emerging next-generation approach [21,22,23,24]. The immune system can also be altered by probiotics which can interact with the mucosa-associated lymphoid tissue directly at immune inductive sites to generate both local mucosal and systemic immune responses [22,23,24,25,26]. *Lactobacillus* spp. specifically exhibit endogenous innate immune-modulating mechanisms including Toll-like receptor (TLR) activation by the cell wall components lipoteichoic acid and peptidoglycan, direct interaction with dendritic cells, and generation of antimicrobial compounds like lactic acid [19,27,28]. *L. casei* expressing porcine VP4 antigen demonstrated proof of concept for *Lactobacillus* spp. as next-generation rotavirus-vaccine vectors, as they induced antigen-specific serum IgG and mucosal IgA, with serum antibodies capable of rotavirus neutralization [29,30].

We have previously demonstrated the feasibility of using *Lactobacillus acidophilus* (LA) as a vaccine candidate by developing recombinant LA (rLA) strains for the mucosal pathogen HIV-1-expressing MPER and Gag epitopes along with IL-1β or the TLR5 ligand *Salmonella enterica* serovar Typhimurium flagellin (FliC) as adjuvants [28,31,32]. Oral immunization with these rLA-generated antigen-specific serum IgG, mucosal IgA, and B-cells in mucosal tissues (female reproductive tract and large intestine). These responses were improved by the adjuvants IL-1β and FliC. We have also established that rLA vaccination neither colonizes the intestine nor permanently disrupts the resident host microbiome [31]. 

To evaluate the potential of our rLA platform as a next-generation rotavirus vaccine, we constructed an rLA rotavirus vaccine strain expressing peptide and protein viral antigens from the VP8 capsid protein [33]. The VP8 capsid protein was selected as the vaccine epitope since some rotavirus-neutralizing antibodies are directed to this protein and the VP8Pep epitope is highly conserved among type-A rotaviruses [34,35]. We also included two immune-stimulating adjuvants, FliC and *E. coli* type-I pilus microfold cell-targeting adhesin protein FimH [32,36]. Adjuvants are essential for mucosal vaccines to overcome the immunotolerant nature of the gut due to high antigen exposure [37]. Here, we present vaccine-strain construction and validation. The immunogenicity of the vaccine was evaluated using a murine in vivo rotavirus challenge model to assess antirotavirus immune responses and protection against a homotypic murine rotavirus. 

## 2. Materials and Methods

### 2.1. rLA Vaccine Strain Construction

(r)LA strains are described in Table 1. NCK56 is a control strain of the human isolate *Lactobacillus acidophilus* NCFM that has plasmid-conferred erythromycin resistance [31,38,39]. VP8-1 and N-terminal FimH integration downstream of the *lba0889* (*enolase*) gene was performed using the *upp*-based counterselective gene-replacement system, as previously described by Douglas et al. [40]. Detailed methods are provided in Appendix A. The VP8Pep epitope was integrated into the *slpA* gene, as previously described [36,41]. The FliC plasmid pTRK1034 was transformed into exponential cultures of NCK2783 by electroporation, as previously described [40,42,43]. *E. coli* and LA strains used to construct the rLA strains are listed in Appendix A [44]. The antigen amino acid sequences and their expression profiles are listed in Appendix A. The primers and plasmids used during cloning are listed in Appendix A and Appendix A, respectively. 

### 2.2. Bacterial Growth Conditions

LA strains were grown anaerobically overnight at 37 °C in MRS media (BD Difco, Franklin Lakes, NJ, USA) supplemented with 5 µg/mL erythromycin when growing antibiotic-resistant strains. The rLA strain GAD85 (Table 1) was grown overnight and then diluted 1:10 to exponential growth. All cultures were washed twice in PBS (Corning, Corning, NY, USA). Colony forming units (CFU) were calculated based on the optical density at 600 nm as previously described [36].

### 2.3. Confirmation of rLA Antigen and Adjuvant Surface Expression by Flow Cytometry 

Resuspended in 100 µL PBS + 1% BSA (Equitech-Bio, Kerrville, TX, USA) + 1:1000 kathon CG/ICP (Supelco, Bellefonte, PA, USA) staining buffer with primary and secondary antibodies was 1 × 10^7^ CFU of (r)LA, as listed in Appendix A. Samples were analyzed on a Gallios flow cytometer (Beckman Coulter, Indianapolis, IN, USA). Analyses were performed in FlowJo (v.10.8.1, Ashland, OR, USA) with gating based on NCK56 and secondary antibody-only controls.

### 2.4. Confirmation of Cytosolic Accumulation of VP8-1 Antigen by SDS-PAGE Western Blot

Pelleted exponential phase cultures were treated with 75 µg/mL lysozyme (Sigma-Aldrich, St. Louis, MO, USA), resuspended in PBS, and incubated for one hour at 37 °C. VP8-1 expressed by *E. coli* and purified by his-tag was included as a positive control. The sample and loading dye (Pierce, Appleton, WI, USA) were incubated at 95 °C for five min and run on a Mini-Protean TGX gel (Bio-Rad, Hercules, CA, USA) in Tris–Glycine–SDS buffer at 200 V. A Precision Plus WesternC ladder (Bio-Rad) was also included. The gel was transferred onto a PVDF membrane and stored overnight at 4 °C in a blocking buffer made of 1% milk + PBST (PBS and 0.05% Tween20) + 1:1000 kathon. The PVDF was incubated with polyclonal mouse anti-VP8 antibody (provided by Dr. Harry Greenberg from Stanford University, Palo Alto, CA, USA) and diluted 1:1000 in blocking buffer. PVDF was washed and incubated with antimouse IgG (clone 7076; Cell Signaling, Danvers, MA, USA) diluted 1:2000 and StrepTactin (Bio-Rad) diluted 1:5000 in the blocking buffer. PVDF was washed and incubated with equal parts of the Clarity Western ECL (Bio-Rad). The membrane was imaged on a Bio-Rad Chemi-Doc XRS+, and lanes were analyzed using Image Lab software (v.6.1; Bio-Rad). 

### 2.5. Animal Ethics and Murine Oral Immunization 

All research involving animals was approved by and in accordance with guidelines set by the Institutional Animal Care and Use Committee of Colorado State University (IACUC #18-1234A). An equal number of male and female rotavirus naïve BALB/cJ mice were obtained from Jackson Laboratories (Bar Harbor, ME, USA). The mice were maintained in specific pathogen-free conditions, housed socially in single-sex groups in commercially available individually ventilated cages, and provided with autoclaved bedding and enrichment. Animals were fed ad libitum commercially irradiated rodent chow (Teklad Global, Envigo, Indianapolis, IN, USA) and tap water filtered via reverse osmosis in autoclaved water bottles. The animals were tracked and monitored daily for any signs of stress or illness. For oral delivery of wild-type LA and rLA strains, washed bacterial cultures were resuspended to 5 × 10^9^ CFU in 200 µL of buffer made of soybean trypsin inhibitor (Sigma-Aldrich) and sodium bicarbonate (NaHCO_3_) [43]. The experimental design is as described in Table 2. The mice were immunized with either the control or the experimental strains for three sequential days every other week (Figure 1, publication license provided in Appendix A). Negative control mice did not receive any intervention. The mice were euthanized by CO_2_ then exsanguination, and tissues were collected at necropsy. 

### 2.6. Murine Rotavirus Challenge 

To generate the wild-type murine rotavirus EDIM-Cambridge (EC_WT_) used for murine infection, 50 µL (400·infectious dose 50; ID_50_) of virus stock (obtained from Dr. Mary Estes at Baylor College of Medicine in Houston, Texas USA) was orally given to 5-day old neonates. Neonates were euthanized 48 h after infection by isoflurane overdose and decapitation. The intestines were collected and tissues pooled, homogenized by bead beating, and diluted in DMEM (Corning) + 1% pen/strep (Hyclone, Logan, UT, USA). Neonates were challenged with serial dilutions of viral stock and screened for the virus in their feces by antigen ELISA to calculate ID_50_ using the Reed and Muench Calculator [45]. Adult mice were orally challenged with a 1 × 10^5^ ID_50_ of EC_WT_ two weeks after the last vaccination (Figure 1). Fecal samples were collected directly before infection, then daily for six days postinfection to detect rotavirus antigen shedding. Mice were euthanized by CO_2_ and then exsanguination seven days after infection. Tissues were collected at necropsy. Fecal pellets were homogenized in PBS + 2X protease inhibitor (ProteaseArrest, G Biosciences, St. Louis, MO, USA), and fecal supernatants were stored at −80 °C following centrifugation for analysis. 

### 2.7. Tissue Collection, Cell Isolation, and Antibody-Secreting Cell (ASC) FluoroSpot Assay 

The spleen and mesenteric lymph node (MLN) were collected in RPMI (Corning), supplemented with 2% l-glutamine (CellGro, Lincoln, NE, USA), 1% HEPES (Caisson Labs, Smithfield, UT, USA), 1% pen/strep (Hyclone), and 0.1% gentamicin (Sigma-Aldrich). Tissues were processed into single-cell suspensions. The spleen was mechanically disrupted using a GentleMACS dissociator (Miltenyi Biotec, Auburn, CA, USA); then, red blood cells were lysed with 7.2 pH ACK lysis buffer (0.15 mol/L NH_4_Cl, 0.001 mol/L KHCO_3_, and 0.0001 mol/L Na_2_EDTA in water). The MLN was mechanically disrupted through a 40 µm cell strainer. Both tissues were centrifuged and then filtered through a 40 µm filter cap tube (Corning). Live cells were counted using a Cellometer Auto2000 (Nexcelom Bioscience, Lawrence, MA, USA). Cells were plated in culture medium of RPMI supplemented with 2% L-glutamine, 2% MEM essential amino acids (Cytiva, Marlborough, MA, USA), 1% pen/strep, 1% HEPES, 1% sodium pyruvate (Cytiva), 1% MEM nonessential amino acids (Cytiva), 5% FBS (Hyclone), 0.1% 2-mercaptoethanol (Thermo Fisher, Waltham, MA, USA), and 0.9% 1 g/L NaOH (Sigma-Aldrich). All media were filter sterilized.

Mouse IgG/IgA double-color FluoroSpot assays (Cellular Technology Limited; CTL, Shaker Heights, OH, USA) were performed following the manufacturer’s instructions. Plates were coated with 2 µg/mL of either VP8-biotin peptide (MASLIYRQLL-biotin, Bio-Synthesis Inc., Lewisville, TX, USA) or VP8-1 (Colorado State University, Fort Collins, CO, USA)-coating antigen. Cell suspensions in culture media from the spleen and MLN collected on the day of the necropsy were added to wells in duplicate at 250,000 cells/well. Plates were developed using 5–10 µL of antimurine IgA (FITC) and IgG (Biotin). Plates were dried overnight, and spots were counted and analyzed using a CTL ImmunoSpot analyzer and associated software (v.7.0.26.0; CTL). To identify CD45^+^CD19^+^ B-cells, 50,000 cells from each tissue on the day of isolation were washed with staining buffer, blocked with antimouse CD16/32 (clone 93; Biolegend, San Diego, CA, USA), and labeled with a cocktail of 7-AAD viability staining solution (Biolegend), FITC antimouse CD45 (clone 30-F11; Biolegend), and Pacific Blue antimouse CD19 (clone 6D5; Biolegend). Flow cytometry was performed using a Gallios flow cytometer (Beckman Coulter, Indianapolis, IN, USA). Analysis was performed using FlowJo (v.10.8.1). Gates were set using fluorescence minus one (FMO) controls after gating for live cells. CD45^+^CD19^+^ B-cell percentages determined from flow cytometry analysis were used to calculate the antigen-specific secreting cells per 1 × 10^6^ B-cells based on the following calculation: $$spotsperwell\cdot (\frac{\left(1\cdot 10^{6}\right)}{cellsperwell\cdot \%B-cellsdeterminedbyflowcytometry})$$

### 2.8. Fecal Antigen Shedding ELISA

Plates were coated with rotavirus polyclonal antibody (Thermo Fisher, PA1-7241) diluted 1:4000 in a 9.6 pH sodium carbonate/bicarbonate buffer (0.015 mol/L Na_2_CO_3_, 0.035 mol/L NaHCO_3_) and incubated overnight at 4 °C. All wash steps were performed five times using PBST unless stated otherwise. Plates were washed and blocked with 250 µL/well of 5% milk + PBS + 1% kathon for three hours, then washed. Undiluted 50 µL/well of homogenized fecal samples were added in duplicate. For plates that included a standard curve to calculate the relative amount of rotavirus antigen shed in murine feces, homogenized intestines of murine neonates infected with EC_WT_ were serially diluted to a 1:1024 dilution in 1% milk + PBS + 0.2% kathon sample diluent. Adult fecal samples were diluted to ensure optical density would fall within this standard curve. Samples were incubated for two hours and plates were washed before adding 50 µL/well of rotavirus NCDV polyclonal antibody HRP (Thermo Fisher, PA1-73015) diluted 1:500 in sample diluent. Plates were incubated for one hour and, after seven washes, 50 µL/well of room temperature (RT) TMB (SeraCare, Milford, MA, USA) were added. The reaction was stopped after 15 min by adding an equal volume of 1 mol/L hydrochloric acid (HCL Thermo Fisher Science Education). The plates were read at a 450–570 nm wavelength on a BioTek (Winooski, VT, USA) Synergy H1 Multimode Reader.

### 2.9. Tissue-Culture-Adapted Murine RV (ETD) Propagation 

The MA104 cell line was obtained from ATCC (CRL-2378.1^TM^) and cultured in complete M199 media (Sigma-Aldrich) supplemented with heat-inactivated 10% fetal calf serum (FCS; heated to 56 °C for 30 min), 1% L-glutamine, and pen/strep (Corning). Incomplete M199 media was only supplemented with 1% L-glutamine and pen/strep without FCS. Cells were passaged using 0.05% 1X trypsin-EDTA (Gibco, Grand Island, NY, USA). Unless specified otherwise, all culturing and incubating conditions were performed at 37 °C and 5% CO_2_; 200 µL of murine ETD rotavirus, a tissue-culture adapted EDIM strain [46], was diluted in incomplete M199 media for an MOI of approximately 0.01. The virus was activated by incubating for 20 min with 2.5 µg/mL of trypsin type IX (Sigma-Aldrich). Confluent MA104 cells were washed in incomplete M199 media. Activated virus was diluted 1 to 5 with incomplete M199 media and incubated for 50 min with the cells. Fifteen mL of incomplete M199 media and 0.5 µg of trypsin/mL were added to the flask, and the infected cells were incubated for 3–4 days until complete cell death. The virus was aliquoted and stored at −80 °C. 

### 2.10. ETD Immunoperoxidase Focus Reduction Neutralization Assay 

The cell-culture-propagated ETD virus was activated with 5 µg/mL of trypsin type IX for 20 min in incomplete M199 media. The virus was diluted in incomplete M199 media to a titer that would form 200–300 foci. In a dilution plate, murine sera were diluted 1:4 in incomplete M199, then serially diluted 1:2 across the plate. Equal volumes of diluted virus and murine sera (60 µL/well total volume) were incubated for one hour in a separate plate. A 96-well plate confluent with MA104 cells was washed twice with incomplete M199 media and 50 µL/well of the virus and sera mixture was incubated for one hour with the MA104 cells. Inoculants were removed, the plate was washed twice with PBS, and 100 µL per well of incomplete M199 media without trypsin was added. The plate was cultured for 16 h, then washed once with PBS. Cells were fixed with 70 µL/well of 10% formalin (37% formaldehyde (Sigma-Aldrich) diluted 1:10 in PBS). The formalin was removed, and the cells were permeabilized with 70 µL/well of 1% Triton-X (diluted in PBS), incubated for three min at RT, and then washed twice with PBS. Seventy µL/well of rabbit antirotavirus hyperimmune serum diluted in PBS + 0.5% BSA was added, and the plates were incubated for one hour at RT. The plate was washed twice with PBS and 70 µL/well of peroxidase-conjugated goat antirabbit IgG (Kirkegaard and Perry Lab, Gaithersberg, MD, USA) in PBS + 0.5% BSA, and the plate was incubated for one hour at RT. The plates were washed with PBS before adding 70 µL/well of AEC substrate (Vector Lab AEC peroxidase substrate kit, Newark, CA, USA). After the color developed for 5–10 min, the reaction was stopped by washing twice with PBS. Foci indicating ETD-infected cells were counted with a microscope. 

### 2.11. ETD-Infected MA104 Cell-Based Antibody ELISA

The virus was activated with 5 µg/mL of trypsin type IX in incomplete M199, then diluted in incomplete M199 media to a titer that would form 200–300 foci. Confluent MA104 cells were washed in incomplete M199 media. Fifty µL/well of diluted virus were added and the plate was incubated for one hour. The virus was removed, plates were washed twice with incomplete M199 media, and then, 100 µL/well of incomplete M199 media were added. The plates were incubated overnight, washed once with PBS, and then cells were fixed with 90 µL/well of 10% formalin for 30 min at RT. Formalin was removed and 90 µL/well of 1% Triton-X (diluted in PBS) were added to permeabilize the cells. Murine sera samples were serially diluted starting at a 1:12.5 dilution in PBS plus 0.2% BSA and 100 µL/well added and incubated for one hour at RT. The plates were washed and 90 µL/well peroxidase-conjugated goat antimouse IgG (Kirkegaard and Perry Lab) in PBS + 0.2% BSA were added to each well and incubated for one hour at RT. The plates were washed with PBS before adding 70 µL/well of AEC substrate. After the color developed for 5–10 min, the reaction was stopped by washing twice with PBS. Positive foci were observed with a microscope with the last titer (1:200 across experimental samples) positive for virus staining considered the titer of the serum sample.

### 2.12. Statistical Analyses

Unless stated otherwise, statistical analyses were performed on macOS in R (v.4.2.2) [47] and RStudio (v.2022.12.0+353) [48] with data visualization performed using the ggplot2 package [49]. To analyze serum antibody titers, a Fisher’s exact test for categorical data was performed to compare the proportions of detectable antirotavirus IgG present across all groups. Titers less than 50 were deemed “not present” and titers equal to or greater than 50 were deemed “present”. A two-way ANOVA followed by Tukey’s HSD to correct for multiple pairwise comparisons was performed for each tissue (spleen, MLN) individually for the ASC assay analyses. The relative amount of EC_WT_ antigen shed in murine feces six days after infection was calculated from a power trendline formula generated in Excel (v.16.73) from the OD values (450–570 nm absorbance) of standard curves from plates run simultaneously. To evaluate differences in the relative amount of EC_WT_ antigen shed between treatment groups, a Shapiro–Wilk test and a Kruskal–Wallis test were performed, then a Dunn’s test for multiple comparisons with Benjamini–Hochburg adjustment per day postinfection to generate adjusted *p*-values. Mice were considered positive for fecal EC_WT_ antigen shedding when raw OD values exceeded a cutoff calculated as the mean plus three times the standard deviation of OD values at day zero across all groups. 

## 3. Results

### 3.1. rLA Expresses Rotavirus Antigens and Mucosal Adjuvants 

The truncated VP8* protein antigen (VP8-1; amino acids 26-231, Appendix A) from the EDIM murine rotavirus strain (GenBank accession AAB94758.2) was double-crossed into the LA genome behind the highly expressed *enolase* gene [50,51]. Cytoplasmic VP8-1 accumulation in rLA was confirmed by Western blot (Figure 2A). The VP8* peptide antigen (VP8Pep; amino acids 1–10, Appendix A) was double-crossed into the surface-layer protein SlpA, as previously described [36].VP8Pep surface expression was confirmed by flow cytometry (Figure 2B). The N-terminal binding domain of the FimH adjuvant was double-crossed into the LA genome in tandem with VP8-1. The FliC adjuvant was expressed from the pTRK1034 plasmid [43]. The surface expression of both adjuvants was confirmed by flow cytometry (Figure 2C,D).

### 3.2. rLA Vaccination Induces Antirotavirus Serum IgG

Mice received six oral immunizations of either control (negative control, buffer, or NCK56) or rLA strains (GAD84 or GAD85) every other week for ten weeks (Figure 1) with the last dose given 24 h before necropsy. Serum was used to measure anti-ETD rotavirus IgG via cell-based ELISA. Mice in both rLA-vaccinated groups had detectable rotavirus-specific IgG while all the control groups did not have a detectable antirotavirus antibody (Figure 3; *p*-value based on a Fisher’s Exact Test comparing proportions across all groups: 0.017). A higher percentage of mice (50%) vaccinated with the GAD85 strain had detectable antirotavirus serum IgG compared to mice vaccinated with GAD84 (12.4%); the rLA nonadjuvanted control strain expressing rotavirus antigens exclusively. One mouse from the GAD85 group exhibited high serum neutralization to ETD with a titer of 800; all other mice across groups had no detectable serum neutralization with titers < 400 (Appendix A). rLA antigen-specific (VP8-1, VP8Pep) fecal ELISAs were also performed, and no fecal IgA was detected. The serum results indicate that rLA vaccination induces a rotavirus-specific antibody response, which is moderately augmented by adjuvants, and there is some potential for eliciting neutralizing antibodies by probiotic vaccination.

### 3.3. Rotavirus Challenge after Vaccination Boosted Antigen-Specific Immunity

There were no antigen-specific (VP8-1 or VP8Pep) ASC responses detected after vaccination in the control or rLA-immunized mice as measured by FluoroSpot. Total IgA and IgG ASCs for the spleen and MLN are provided in Appendix A, respectively. To evaluate the impact of the rotavirus challenge on antibody-secreting CD45^+^CD19^+^ B-cell populations, the mice were orally dosed with either buffer, NCK56, or GAD85 every other week over eight weeks. Two weeks after the final dose, all groups were challenged with 1 × 10^5^ ID_50_ of EC_WT_ rotavirus. Fecal samples were collected daily for six days after infection and tissues for ASC assays were collected at necropsy seven days post rotavirus challenge. Antigen-specific (VP8-1, VP8Pep) cells per 100,000 antibody-secreting (IgA, IgG) CD45^+^CD19^+^ B-cells were measured in the spleen (Figure 4A) and MLN (Figure 4B). Insignificant (adjusted *p*-value: <0.05) comparisons between groups share a common letter. GAD85-vaccinated mice had significantly more antigen-specific splenic antibody-secreting B-cells compared to the buffer and NCK56 control groups for both VP8-1-specific IgA and IgG and VP8Pep IgA (Figure 4A; all adjusted *p*-values were <0.05). More VP8Pep IgG-secreting cells were detectable in GAD85 compared to the control groups, but the difference was not significant (all adjusted *p*-values: >0.05). 

GAD85-vaccinated mice generated a VP8-1-specific IgG response in the MLN, which was significantly increased compared to both controls (all adjusted *p*-values: <0.05). The remaining comparisons were found not to be significant between all groups (all adjusted *p*-values: >0.05). The NCK56 group had detectable but insignificant antigen-specific responses in the spleen and MLN compared to GAD85, and the buffer group had no detectable antigen-specific B-cell responses in either tissue (all adjusted *p*-values: >0.05). No significant differences were detected between the NCK56 and buffer groups in either tissue (all adjusted *p*-values: >0.05). Adjusted *p*-values for the spleen and MLN are provided in Appendix A, respectively. Antigen-specific secreting B-cell populations were boosted after the EC_WT_ viral challenge, and rLA vaccination was able to generate B-cell responses specific to both antigens. The full-length protein was the sole antigen with specific B-cell responses across tissues. 

### 3.4. rLA Vaccination Delays Rotavirus Shedding 

The relative amount of rotavirus antigen shed in feces was quantified by ELISA for six days following the EC_WT_ viral challenge (Figure 5A). A Shapiro–Wilk test determined that the data were not normally distributed, and a Kruskal–Wallis test indicated significant differences were detectable between groups (*p*-values: <0.0001; *p*-values provided in Appendix A). Mice vaccinated with GAD85 shed significantly less virus a day following infection compared to both control groups (adjusted *p*-values: 0.029 for buffer and 0.0002 for NCK56) with a 4864 average relative amount of EC_WT_ antigen for GAD85-vaccinated mice, with 147,075 and 429,761 averages for the buffer and NCK56 groups, respectively. Averages rounded to the nearest whole number are provided in Appendix A and adjusted *p*-values in Appendix A. The percentage of mice positive for EC_WT_ antigen shedding (Figure 5B) was determined based on a cutoff three times the standard deviation plus the mean of day zero OD values across groups. Twenty-five percent of GAD85-vaccinated animals were deemed positive for rotavirus antigen shedding on day one postinfection, while 100% in both of the control groups were positive. These results suggest that rLA vaccination can provide some reduction of EC_WT_ rotavirus shedding in vivo.

## 4. Discussion

There is a clear and significant need for next-generation alternatives to the currently available attenuated live human rotavirus vaccines that can address the relatively low efficacy in LMIC [20,35]. The probiotic strain *Lactobacillus acidophilus* is an attractive vaccine candidate due to its safety profile, ability to tolerate the acidic stomach environment, and endogenous interaction with the mucosal immune system [26,28]. In this study, we constructed a novel recombinant LA rotavirus vaccine that expresses two rotavirus antigens and two exogenous adjuvants, then assessed immunogenicity and efficacy after oral immunization and viral challenge. This is the first rLA rotavirus platform, to the authors’ knowledge, that delivers multiple viral antigens and mucosal adjuvants in combination. 

Vaccination with rLA-induced VP8 antigen-specific ASC and whole rotavirus-specific IgG immune responses. A greater percentage of GAD85-immunized mice were seropositive for rotavirus compared to mice in the GAD84 nonadjuvanted control group. This suggests that the adjuvants FliC and FimH are potentially important for rLA immunogenicity, resembling the established role of adjuvants described for other mucosal vaccines [28,52]. Although antigen-specific ASCs were not detected in the control or the rLA groups following vaccination, mice vaccinated with GAD85 exhibited ASC responses specific to both the rotavirus VP8 peptide and protein antigens in systemic (spleen) and local (MLN) lymphoid tissues seven days after viral challenge with EC_WT_. This indicates that memory B-cells were primed during vaccination and the immune response was activated after rotavirus infection. It was surprising that antigen-specific ASC responses were not detected after vaccination alone. It has been demonstrated that the detection of rotavirus-specific memory B-cells in tissues like the spleen may require ongoing antigen stimulation [14]. We have previously shown that recombinant probiotics are cleared from the gastrointestinal tract by 72 h postdelivery [53]. Therefore, rotavirus antigen would no longer be present in tissues two weeks after the final full vaccination, making it likely the levels of rotavirus-specific ASC in the spleen and MLN would be low and possibly below the level of detection of the FluoroSpot assay. The levels of ASC we detected after the challenge are comparable to those seen in other animal models [54,55]. Additionally, antigen-specific fecal IgA was not detected by ELISA. It should be noted that these results were difficult to interpret due to the high background and limited sample availability. BALB/c mice are known to have abundant polyreactive IgA antibodies, which can obscure the detection of vaccine-induced antigen-specific responses [56]. The presence of the antigen-specific IgA B-cells, indicating IgA induction, on FluoroSpot is a reasonable proxy for fecal IgA detection by ELISA. Additional studies are required to phenotypically characterize rotavirus-specific B-cell populations in lymphoid tissues and effector sites, particularly the intestinal lamina propria [14,25].

In this study, mice were immunized with an rLA vaccine expressing antigens from the VP8* capsid protein sequence from the EDIM rotavirus strain (G3[P16] genotype) and were challenged with the homotypic yet different EC_WT_ strain (G3[P16]) [57,58]. We saw a one-day delay of fecal viral antigen shedding in the GAD85-immunized group, indicating the potential of our rLA construct to generate modest, though neither complete nor robust, protective immune responses against other homotypic rotaviruses. Current rotavirus vaccines are generally protective against both monotypic and heterotypic circulating rotavirus strains yet offer slightly less protection for partially or fully heterotypic strains [59,60]. It is likely that alternative or combination antigens other than VP8* alone may prove more immunogenic and ultimately protective. This is supported by the early closure of a clinical trial evaluating the efficacy of a P2-VP8 subunit injectable vaccine that did not provide improved protection against severe rotavirus diarrhea compared to the currently approved oral vaccines (ClinicalTrials.gov ID NCT04010448) [61,62,63]. Although supplementation with wild-type probiotic strains including LA NCFM has been shown to delay rotavirus shedding in animal models and human infants [64,65], this was not observed in the analogous NCK56 group. This may be due to the timing of vaccination since mice in this study received their final dose of the NCK56 wild-type control strain two weeks before the challenge. 

A limitation of our study is an inability to assess protection from clinical disease, as we used an adult murine model. Adult mice do not display clinical symptoms of rotavirus infection (e.g., presence and prolonged duration of diarrhea and pathology characteristic of rotavirus) [24]. We were therefore restricted to using rotaviral antigen shedding as a proxy for measuring rLA-induced protection from disease. Nevertheless, BALB/c mice are a conventional small-animal model of rotavirus infection, as neonates have been studied since 1947 and the adult model was established in 1990 [66,67]. Studying adaptive immunity is critical for vaccine development and can be more readily evaluated in adult mice as neonates must be infected within approximately 10 days of birth to observe clinical disease [66]. Assessing the rLA rotavirus vaccine’s potential to abate morbidity and mortality is necessary. To this end, using a neonatal pig model is a logical next step in our vaccine development, as they show clinical symptoms and can be infected with human rotavirus strains [15,24]. 

## 5. Conclusions

This study highlights the promise of engineering the probiotic *Lactobacillus acidophilus* as a next-generation rotavirus vaccine to help overcome barriers against mucosal immunization. Future studies using relevant animal models will determine if this oral vaccine approach can address the reduced effectiveness observed in LMIC with current rotavirus vaccines [2].

## 6. Patents

R.B., G.A.D., and A.C.V. are inventors on a patent application related to the engineering of probiotics for vaccination purposes. 

## Figures and Tables

**Figure 1 vaccines-11-01774-f001:**
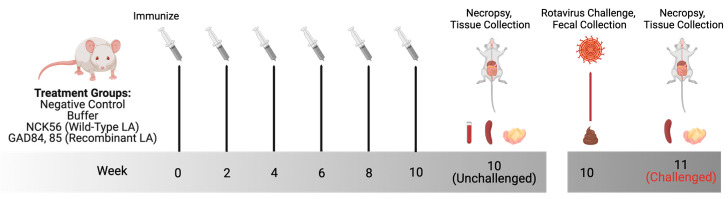
Experimental design. To evaluate host response to the vaccination, mice were vaccinated by oral gavage every two weeks with either buffer (*n* = 8), the wild-type *Lactobacillus acidophilus* (LA) strain (NCK56; *n* = 8), dual-antigen recombinant *Lactobacillus acidophilus* (rLA) strain (GAD84; *n* = 8), or dual-antigen/dual-adjuvant rLA strain (GAD85; *n* = 8). Negative control (*n* = 7) mice received no intervention. Mice were administered a sixth vaccination dose a day before the necropsy. Spleen, mesenteric lymph node (MLN), and sera were collected at necropsy. To determine host response to the challenge, mice were orally dosed with either buffer (*n* = 8), NCK56 (*n* = 8), or GAD85 (*n* = 8) for eight weeks before infection with EC_WT_ two weeks after the final immunization. Fecal samples were collected daily for six days following infection with spleens and MLN collected at necropsy on day seven. Figure created with BioRender.com.

**Figure 2 vaccines-11-01774-f002:**
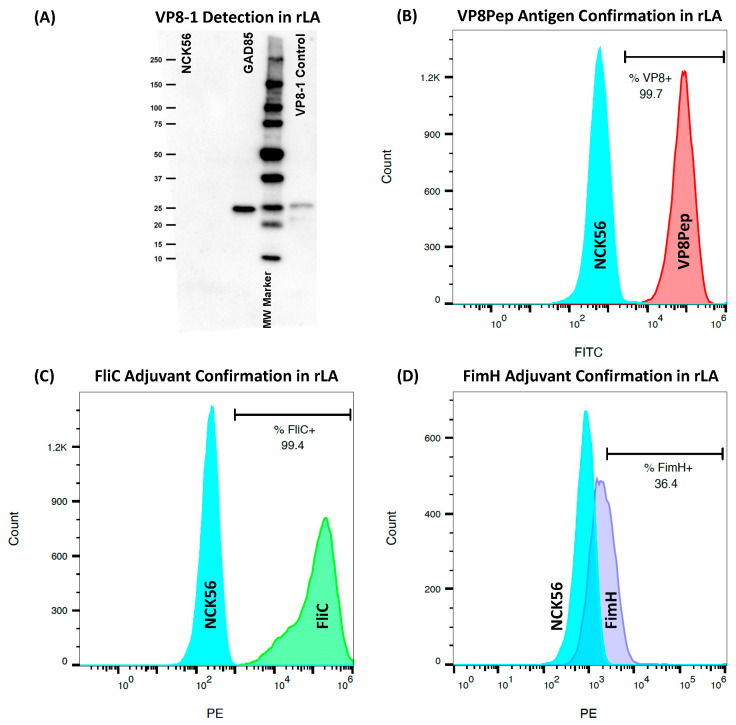
Rotavirus rLA vaccine-strain confirmation. Western blot (**A**) of VP8-1 antigen protein (25 kDa, as measured by the molecular weight marker) exclusively present in GAD85 lysate corresponding to the *E. coli* expressed positive VP8-1 control. No detectable band was observed in the NCK56 lysate negative control. The uncropped Western blot for GAD85 is provided in Appendix A and the densitometry readings are in Appendix A. Flow cytometry histograms confirming surface expression of (**B**) VP8Pep, (**C**) FliC, and (**D**) FimH for rLA constructs compared to the NCK56 control (blue peaks). Confirmation of dual-antigen expression for the rLA strain GAD84 is provided in Appendix A (uncropped Western blot) and Appendix A (flow cytometry histograms) with the densitometry readings in Appendix A. VP8-1: 206 amino acid rotavirus protein antigen, VP8Pep: VP8 10 amino acid rotavirus peptide antigen, FliC: flagellin adjuvant, FimH: adhesion adjuvant.

**Figure 3 vaccines-11-01774-f003:**
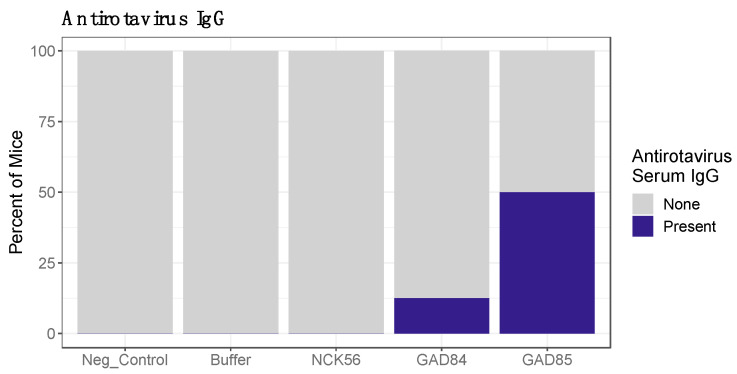
Antirotavirus serum IgG titers in the following vaccination. Mice in the negative control, buffer, and NCK56 groups did not have a detectable serum antibody response to the ETD rotavirus strain. A higher percent of mice vaccinated with the rLA strain expressing both antigens and adjuvants (GAD85) had an antiviral IgG response present compared to the rLA strain only expressing antigens (GAD84).

**Figure 4 vaccines-11-01774-f004:**
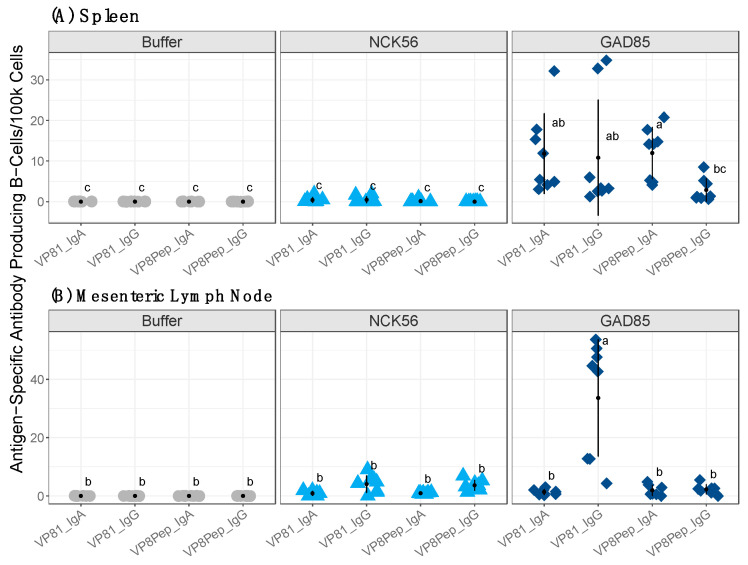
Antigen-specific antibody-producing cells per 100,000 CD45^+^CD19^+^ B-cells as measured by FluoroSpot in the (**A**) spleen and (**B**) MLN. Insignificant comparisons between groups share a common letter (a, b, or c). GAD85 had significantly more antigen-specific B-cell responses compared to both controls in the (**A**) spleen. The VP8Pep-specific IgG trended higher in the GAD85 group, but this difference was not significant. VP8-1-specific IgG was the only antigen-specific response significantly increased in GAD85-vaccinated mice compared to both control groups in the (**B**) MLN. All other responses were not significantly different between groups. The buffer and NCK56 did not differ significantly from one another across tissues. Significant adjusted *p*-values are <0.05 and insignificant are >0.05 based on a two-way ANOVA followed by Tukey’s HSD. MLN: mesenteric lymph node.

**Figure 5 vaccines-11-01774-f005:**
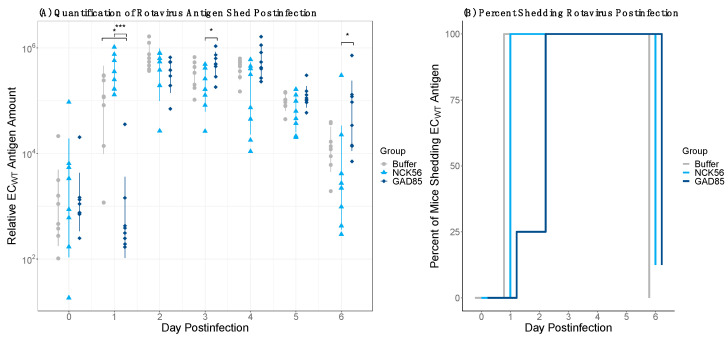
Rotavirus EC_WT_ fecal shedding following oral rotavirus challenge. (**A**) Relative amount of EC_WT_ rotavirus antigen shed in fecal samples for six days after viral infection as quantified by ELISA using a standard curve. There is a significant, one-day delay in antigen shedding in the GAD85 group compared to the buffer and NCK56 groups. Statistical analyses were performed on untransformed data, and the *y*-axis is visualized on a log-10 transformed scale. (**B**) All mice from the buffer and NCK56 groups were positive for antigen shedding by day one, with only 25% of the GAD85 group shedding virus. One hundred percent of mice across groups were shedding virus by day two; 87.5–100% of mice were negative for viral shedding by day six across groups. Adjusted *p*-value: ‘***’ < 0.001, ‘*’ < 0.05 based on a Dunn’s test with Benjamini–Hochburg adjustment.

**Table 1 vaccines-11-01774-t001:** Wild-type *Lactobacillus acidophilus* (LA) and recombinant LA (rLA) strains used for oral immunization in this study. Erythromycin: Erm.

Strain	Components	Resistance	Reference
NCK56	Wild-type lab LA NCFM strain	Erm	[38]
GAD84	LA-expressing VP8Pep within SlpA and chromosomal VP8-1 downstream of *enolase*	None	This study
GAD85	LA-expressing VP8Pep within SlpA, pFliC, and chromosomal VP8-1 and FimH downstream of *enolase*	Erm	This study

**Table 2 vaccines-11-01774-t002:** Study design describing treatments used in vaccination and challenge experiments.

Group	N per Treatment	Treatment	Challenged with Rotavirus
Negative Control	7	No Intervention	No
Buffer	8	Bacterial Resuspension Media	Yes
NCK56	8	Wild-type LA Control	Yes
GAD84	8	Dual-Antigen rLA	No
GAD85	8	Dual-Antigen and Dual-Adjuvant rLA	Yes

## Data Availability

The data is contained within the article or the Appendix A, which can be downloaded at: www.mdpi.com/article/10.3390/vaccines11121774/s1.

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
