# Peer review of "Lactobacillus acidophilus Expressing Murine Rotavirus VP8 and Mucosal Adjuvants Induce Virus-Specific Immune Responses"

_vaccines, 2023, doi:10.3390/vaccines11121774_

Round 1

Reviewer 1 Report

Comments and Suggestions for Authors

In recent years, important efforts have been made to design new-generation rotavirus (RV) vaccines that work efficiently in low- and middle-income countries. Thus, it is important to study conserved target antigens that induce heterosubtypic responses. In this study, the authors explore the possibility of using Lactobacillus acidophilus (LA) as a vector to induce a protective immune response against RV infection in a murine model. As antiges models, a truncated form of RV VP8  from the mouse RV EDIM  comprising aa 26-231 (VP8-1) and the 10 aa peptide from the VP8 sequence (VP8pep), target of neutralizing Abs, were used. VP8 was selected because there is some evidence that it can induce protective heterotypic responses. VP8-1 and VP8pep were expressed on rLA, the first intracellularly and the second on the membrane, together with the adjuvants pFliC and FimH that were expressed in the membrane (GAD85). A rLA expressing VP8-1 and VP8pep (GAD84) and a non-recombinant LA (NCK1910) were used as controls.

Mice were immunized orally six times with GAD85, GAD84, NCK1910, or buffer. Before the virus challenge, the serum and cells from the spleen and mesenteric lymph node (MLN) were obtained. It was found that only GAD85 and GAD84 induced anti-VP8 serum Abs, and that GAD85 showed higher induction levels. Only GAD85 induced neutralizing serum Abs. It was shown that the adjuvants were important for an optimal serum Ab response against VP8. No memory B cells against VP8-1 or VP8pep were found in the cells from the spleen or MLN.

When mice were challenged with the murine RV EC only the mice treated with GAD85 showed a reduction of the viral load in feces on day one. Interestingly, it was found that cells from the spleen and MLN had memory B cells against VP8-1 and VP8pep. The authors concluded that the vaccination with GAD85 can induce an immune response against VP8 that can protect the mice against an RV infection in a heterotypic way. The manuscript is clear and well-written. However, major issues must be addressed to support the study's main conclusion.

Comments:

1.- Why only in GAD85 and not in GAD84 the expression of VP8-1 and VP8pep was tested by flow cytometry?

2.- The main conclusion is that the immune response induced by GAD85 is protective against an RV infection. However, after six oral inoculations of GAD85 there was only a reduction of the virus load in feces in the first 24 hrs after the viral challenge. During the rest of the seven days, the viral excretion levels in the feces of the controls and GAD85 were similar. With these data, it is difficult to conclude that GAD85 induces protection.

3.- The authors claim that the protection against the infection is heterotypic. VP8-1 and VP8pep are from the reported sequence of the murine RV EDIM that is G3[P16], and the virus used for challenging the mice was the murine RV EC that has the same G serotype and P genotype; therefore, the response is homotypic.

4.- Intestinal IgA, but not serum Abs, is an important correlate of protection against the RV infection. However, the authors did not evaluate the levels of anti-VP8 specific IgA Abs in the feces of the infected mice. It is essential to assess this variable.

5.- It is difficult to explain why there were no detectable memory B cells specific for VP8-1 and VP8pep in the spleen and MLN after the immunizations with GAD85 before the viral challenge. Still, they were detectable only after the challenge. It should be discussed in more detail.

Reviewer 2 Report

Comments and Suggestions for Authors

The authors present a very interesting and well described study. The presented data support the conclusions drawn.

I did not identify any issues with this study or the manuscript.

Line 51 suggest revision "in immune suppressed populations"

There are several instances where sentences begin with numbers. The authors could consider revising these, for example, line 120, in fitting with writing conventions.

Reviewer 3 Report

Comments and Suggestions for Authors

Darby Gilfillan and colleagues (vaccines-2577649) presented an investigation on novel recombinant Lactobacillus acidophilus (rLA) vaccine expressing rotavirus antigens of the VP8* domain from the rotavirus EDIM VP4 capsid protein along with the adjuvants FimH and FliC. It is a informative and well organized work, and I have a few minor points.

1. a short description about LA control (NCK56) should be added.

2. it remains unknown how GAD84 and GAD85 are designed and, more importantly, how technically was created.

3. "Equal number male and female... were obtained" and in single sex groups, how the sex number were chosen in each groups (five groups in table 2)

4. Figure 5, there is no difference between NCK56 and GAD85. can you explain that.

5. I am wondering an infection with fatal by murine rotavirus, that you can test the protection efficiency for different LAB variant.

Comments on the Quality of English Language

It is a well written manuscript.

Reviewer 4 Report

Comments and Suggestions for Authors

Gilfillan, Vilander et al. aim to develope a probiotic-based vaccine against rotavirus. 

They used Lactobacillus acidophilus (LA) as an expression vector to express mouse rotavirus protein VP8 or a short VP8 peptide (VP8p 10 amino-acids) with (GAD85) or without (GAD 84) mucosal adjuvant proteins (FimH and FliC). Expression of VP8, VP8p and FimH and FliC was controlled. Adult mice were immunized with these recombinant LA strains (control being injected with the non-recombinant LA or buffer).  The immune response was evaluated by measuring the anti-rotavirus antibodies, the serum neutralization titer and, after a challenge with another mouse rotavirus (ECwt), number of antibody secreting cells in spleen and lymph node and rotavirus shedding in feces were measured. 

The presence of anti-rotavirus IgG (only 50% of the GAD85 (VP8+adjuvant) group developed an IgG response against rotavirus but only one (out of 8) serum was neutralizing.

Following ECwt challenge,

1-    GAD85-vaccinated mice had significantly more antigen-specific B-cell responses in the spleen compared to controls

2-    VP8-1-specific IgG was the only antigen-specific response significantly increased in GAD85-vaccinated mice compared to both control groups in the mesenteric lymph node.

3-    GAD85-vaccinated mice shed significantly less virus a day following infection compared to both control groups. Twenty-five (2/8) per cent of GAD85-vaccinated animals were shedding rotavirus antigen day one post-infection while 100 % in both the control groups were positive.

These immunological responses are quite modest but encouraging.

The experiments are well conducted with appropriate controls. The paper is overall well written but requires some rephrasing (see bellow).

Figure 2 A: the VP8 band from GAD85 seems to be shorter than the rVP8 from bacteria. Please explain.

Table 2:is not clear; “treatment” is indicated by two lines not aligned with “group”; it is not clear at first glance; please add blank lines between groups and align treatments and groups. For clarity, GAD84 could be labelled “dual-antigen rLA “.

Line 117 “Colony forming units (CFU) were calculated using an optical density of 600nm”.  Not clear.

Line 134 and 224 « kathon » (with a capital letter?) is a brand name for several products; the exact composition or the exact product’ name should be provided.

Line 246-248. “Activated virus was diluted 1 to 5 with incomplete M199 media in a T75 flask and incubated for 50 min. Activated virus was diluted another 6X with incomplete M199 media and 0.5 μg of trypsin/mL was added.” Is it that 1ml of virus was used to infect cells in a T75 flask in a final volume of 5ml then after 50mn 25ml of medium plus trypsin was added? please make it clearer and give an (at least) approximated multiplicity of infection (MOI).

Line 254 “titer that would form 200-300 foci.”  A titer is foci/volume.

Line 260 media with 100 μl/mL of trypsin added. Please give the final concentration of trypsin in μg/ml.

Line 286 “Positive foci were observed with a microscope with the last titer positive for  virus staining considered titer of serum sample.” Titer of the last positive well?

Comments on the Quality of English Language

none

Reviewer 5 Report

Comments and Suggestions for Authors

Rotavirus diarrheal sickness continues to be a leading cause of death in children under the age of five, currently, 125 countries introduced rotavirus vaccine into national immunization program (https://view-hub.org/vaccine/rota). However, current rotavirus vaccines, which are attenuated live viruses, have limited efficacy in low and middle-income countries, leaving many children vulnerable to infection. Furthermore, attenuated vaccines have inherent risks, such as potential virulence reversion and recombination with circulating viruses. To overcome these challenges, the authors this manuscript constructed a new recombinant Lactobacillus acidophilus (rLA) vaccine candidate that expresses rotavirus antigens from the VP8* domain of the rotavirus EDIM VP4 capsid protein, as well as two immune-stimulating adjuvants, FimH and FliC.  After the confirming VP8 epitope and adjuvant expression with flow cytometry and western blot analysis, they orally vaccinated adult BALB/cJ mice with the new probiotic vaccine candidate, followed by the challenge the mice with the murine rotavirus strain ECWT. The result showed that rLA-vaccinated mice produced anti-rotavirus serum IgG and exhibited antigen-specific antibody secreting cell responses. However, fecal antigen shedding in the rLA group was only lower than control groups on the first day after challenge, and it remained similar or higher than that in other groups later on. These findings suggest that novel rLA constructions can successfully elicit immune response against rotavirus, though the heterotypic protection against rotavirus challenge needs further investigation. Overall, the manuscript is well-written, and the study highlights the possibility of a probiotic next-generation vaccine construct against human rotavirus. There are a few points that need to be addressed.

1.      In the introduction section, lines 39 to 40, the data has been updated, please cite the latest information (https://view-hub.org/vaccine/rota).

2.      The experiment design shown in Fig 1 is not clear enough. For example, the time of necropsy, the time points of 10 and 11 weeks should be separated. Also, why did the authors necropsy the challenged and unchallenged animal at different time points?

3.      Fig 2a, some bands of marker were highlighted by imaging system, you may want to uncheck the “Highlight saturated” option in imaging system. 4.      The polyclonal antibody was used as capture antibody in the Fecal antigen shedding ELISA, and vaccine candidate GAD85 expresses VP8.  It would be interesting to know whether the ELISA can detect GAD85 in fecal samples. Therefore, it would be ideal to include a nonchallenge GAD85 immunized control for the animal study.

Round 2

Reviewer 1 Report

Comments and Suggestions for Authors

The authors addressed all the concerns in the article. I recommend the publication of this new version of the manuscript.

Reviewer 5 Report

Comments and Suggestions for Authors

All of the reviewers' comments were addressed by the authors. Accept for publication is my recommendation.